# Relationship between the Consumption of Fermented Red Beetroot Juice and Levels of Perfluoroalkyl Substances in the Human Body’s Fluids and Blood Parameters

**DOI:** 10.3390/ijms241813956

**Published:** 2023-09-11

**Authors:** Magdalena Surma, Tomasz Sawicki, Mariusz Piskuła, Wiesław Wiczkowski

**Affiliations:** 1Malopolska Centre of Food Monitoring, Faculty of Food Technology, University of Agriculture in Krakow, 122 Balicka St., 30-149 Krakow, Poland; 2Department of Human Nutrition, Faculty of Food Sciences, University of Warmia and Mazury in Olsztyn, Słoneczna 45F St., 10-719 Olsztyn, Poland; 3Institute of Animal Reproduction and Food Research, Polish Academy of Science, Tuwima 10 St., 10-748 Olsztyn, Poland

**Keywords:** perfluoroalkyl substances, fermented red beetroot juice, human body fluids, blood parameters

## Abstract

Per- and polyfluoroalkyl substances (PFASs) are a group of fluorinated, organic, man-made chemicals; they do not occur naturally in the environment. This study aimed to determine the profile and content of PFASs in the volunteers’ blood plasma and urine after the consumption of fermented red beetroot juice and then correlated it with the blood parameters. Over 42 days, 24 healthy volunteers ingested 200 mL/60 kg of body weight of fermented red beetroot juice. PFASs were analyzed using the micro-HPLC-MS/MS method. Five perfluoroalkyl substances were found in the volunteers’ body fluids. After consuming the juice, it was discovered that regarding the perfluorocarboxylic acids, a downward trend was observed, while regarding the perfluoroalkane sulfonates, and their plasma content showed a statistically significant upward trend. Analysis of the hematology parameters indicated that the intake of fermented red beetroot juice showed a significant decrease in mean corpuscular volume (MCV), platelets concentration, mean platelet volume (MPV), platelet large cell ratio (P-LCR) at the significance level *p* < 0.01, and hematocrit (*p* < 0.05). On the other hand, the dietary intervention also indicated a significant (*p* < 0.01) increase in corpuscular/cellular hemoglobin concentration (MCHC). In the case of blood biochemistry, no significant change was observed in the blood samples after the intake of the fermented beetroot juice. However, a decreasing tendency of total cholesterol and low-density lipoprotein concentration (LDL-C) was observed. Based on the presented results, there is a need to analyze and monitor health-promoting food regarding undesirable substances and their impact on consumer health.

## 1. Introduction

Nowadays, it is believed that vegetables and the nutrients obtained from them are a crucial part of the daily human diet [1]. These products are considered to be among the best sources of essential nutrients and bioactive compounds [2]. In addition, many studies suggest that the intake of vegetables positively affects consumer health. The regular consumption of vegetables can positively affect the prevention of many diseases, such as cancer (colorectal and breast cancer) and cardiovascular diseases [3,4,5].

Red beetroot (*Beta vulgaris* L. subsp. *vulgaris*) is one of the most popular vegetables as a valuable source of vitamins, minerals, and bioactive compounds [6]. Due to the high phytochemical content, the intake of this vegetable may have positive, beneficial effects for consumers. Red beetroot shows anti-neurodegenerative, antitumor, anti-inflammatory, antibacterial, antiviral, cardioprotective, and lipid peroxidation inhibitory activities [7]. Moreover, this vegetable is trendy in the food industry. It produces many red beetroot products or additives (such as drinks and concentrated juices, yoghurts, and frozen foods) and is used in the preparation of natural red dyes [6]. On the other hand, red beetroot may contain other compounds, such as perfluoroalkyl substances (PFASs) [2].

These molecules are a wide range of synthetic organofluorine compounds which have been used commonly in the industry [8]. They are a diverse group of compounds with high thermal, chemical, and biological inertness [9,10]. PFASs remain stable in the presence of acids, bases, oxidants, and reductants. Furthermore, they exhibit high resistance to degradation through photolytic or metabolic processes and microbial decomposition. Nevertheless, their widespread use has led to the contamination of the environment, which, as a consequence, has led to the exposure of these compounds in humans and animals [8,11]. They are a group of organofluorine compounds, aliphatic hydrocarbons, in which all or almost all hydrogen atoms are replaced with fluorine. They consist of a hydrophobic alkyl chain of varying lengths (typically C4 to C16) and a hydrophilic end group, which may be fully or partially fluorinated. PFASs are classified as persistent and bioaccumulative substances [12]. PFAAs are ubiquitous in various environmental media and are distributed globally. Due to their ability to migrate, they can be transferred from water to soils, taken up by plants, and thus enter the food chain. These substances enter the human body through the digestive and respiratory systems, as well as through the skin. Moreover, they are not metabolized and accumulate in the body [13]. Furthermore, direct and indirect contact with PFAS-containing materials such as oil- and water-resistant materials, detergents, paints, and fabrics causes a high exposure of humans to their toxic properties [10]. In living organisms, high absorption levels and low elimination rates of PFASs have been observed [14]. 

Researchers confirmed that PFASs can be present in human and animal tissues, and in blood (serum or plasma). Studies indicated that anionic PFASs were detected primarily in the human plasma/serum [15]. The biomonitoring of PFASs in the human body started in 2000. Many PFASs have been detected in human matrices, most commonly in blood samples [16,17]. In recent years, a number of papers have confirmed the occurrence of these compounds in human blood (both in serum and plasma), as well as in the umbilical cord and maternal blood [8,17,18,19,20,21,22,23]. In addition to blood, some PFASs have also been found in other human tissues. PFASs were predominantly found in the liver [24,25]. Furthermore, studies show some accumulation potential in the lungs, kidneys, bones, and the brain [26]. Numerous recent studies have confirmed their presence in seminal plasma [27], the breast milk [28] of lactating mothers, and umbilical cord blood, all drawing attention to their influence on the human reproductive system. However, in contrast to most other persistent organic pollutants (POPs), they do not tend to accumulate in fat tissues but bind to serum albumin and other cytosolic proteins and accumulate mainly in the liver, the kidneys, and in bile secretion [26,29]. In addition, positive associations were observed between levels of PFOS and PFOA detected in the serum, and total cholesterol, low-density lipoprotein cholesterol, and thyroxine and thyroid hormone concentrations [30,31].

In conjunction with the bioaccumulation and biomagnification potential of PFASs, these long half-lives can give rise to various processes within the living cell and lead to concern over their potential hazard to human health. Because of their capacity to modify surface properties, even at the molecular level, it is essential to elucidate their toxicity and toxicokinetic activity [32]. Both PFOA and PFOS have shown moderate acute toxicity via ingestion. The oral LD50 levels assessed for PFOS were 230 and 270 mg kg^−1^ bw for male and female rats, respectively [33]. In contrast to PFOS, PFOA is moderately toxic. The LD50 value in rats ranged from 430 to 680 mg kg^−1^ bw [34]. The suspected toxic effects of PFASs include the following: liver toxicity, including liver hypertrophy; liver cancer; disruption of lipid metabolism due to their effect on serum cholesterol and triglyceride levels; function of the immune system, causing atrophy of the thymus and spleen or suppressed antibody responses; function of the endocrine system due to their effects on thyroid hormone levels (triiodothyronine (T3) and thyroxine (T4)); induction of adverse neurobehavioral reactions; tumor formation; prenatal and neonatal toxicity; decreased birth weight and size; and even obesity [16,35,36,37,38,39,40,41,42,43,44,45,46]. Several studies cover the putative modes of action of PFASs on a cellular level, but these mechanisms still need to be fully defined. Nevertheless, due to the structural similarities of PFASs to endogenous fatty acids, these reactions can be partly attributed to their morphology, and more precisely to activity resulting from their chemical structure. PFASs are characterized by the high tendency to noncovalent, intracellular binding to β-lipoproteins, albumin, and other plasma proteins, such as fatty acid-binding protein (L-FABP) [25]. The proliferation of peroxisomes is one of the main reasons for liver toxicity observed in laboratory animal studies.

PFASs are extensively applied in various industrial and consumer goods. Fluorosurfactants are more effective and efficient in surface activity than their hydrogenated analogues. These and other properties, such as high thermal, chemical, and biological stabilities, make PFASs a perfect material for industrial and domestic applications [47,48]. Typical applications encompass the automotive and aviation industries (hydraulic fluids, low-friction bearings and seals, and materials for car interiors), construction technology (paints and coating additives and glues), biocides (herbicides and pesticides), electronics (flame retardants, weather resistant coatings, and insulators), household products (wetting and cleaning agents, nonstick coatings, and components of cosmetic formulations), medical articles (stain and water repellents in surgical equipment, raw materials for implants), and packaging materials (oil and grease repellent materials) and textiles (impregnating agents for fabrics, leather, and breathable membranes) [10,49,50].

Due to the presence of PFAAs in ambient air, various consumer products, drinking water, and food, it has become necessary to assess their potential impact on human health accurately. Researchers have reviewed the occurrence of highly fluorinated compounds in human matrices in recent years, and there is indisputable evidence that their bioaccumulation potential in tissues is high [51,52]. However, there is uncertainty regarding the accumulation processes themselves and the acute or chronic toxicity effects due to variations in observed toxic response to PFAAs between species and genders within tested species. Considering all the aspects, this study aimed to determine the perfluoroalkyl substances’ content in fermented red beetroot juice and human body fluids (blood plasma and urine) before and after intake of this product. The novelty of the work lies in linking dietary exposure to PFASs and blood parameters, bringing a new perspective to our understanding of these compounds. In this study, we wanted to observe whether, apart from compounds with beneficial properties for health, this product is a source of toxic substances and whether they can affect the biomarkers of the human body. In the case of the effect of the fermentation process on the PFAS content and bioavailability, there is no such scientific data. To our best knowledge, this paper is the first detailed and cross-sectional investigation of selected perfluoroalkyl substances in human body fluids after exposure to fermented red beetroot juice.

## 2. Results and Discussion

### 2.1. Perfluoroalkyl Substances Content

The content of perfluoroalkyl substances was tested in fermented red beetroot juice and the human body fluids (blood plasma and urine) of volunteers before and after the intake of these products (Table 1). In urine samples, the PFOA was the only perfluoroalkyl substance found in trace amounts (<LOQ). This confirms the information found in the literature, i.e., that these compounds metabolize very poorly, and thus they are excreted from the human body very poorly; they only accumulate in it, which makes them even more dangerous [13]. For PFASs, a high absorption level is observed with a low elimination rate of the substance [14]. Available epidemiological studies support an association between exposure to certain PFASs and various health outcomes, including altered immune and thyroid function, liver disease, lipid and insulin dysregulation, kidney disease, adverse reproductive and developmental effects, and cancer [53].

In the tested fermented red beetroot juice, among the analyzed PFASs, only the most common perfluoroalkyl substance apart from the PFOS, i.e., the PFOA, was identified, the content of which was 0.099 ± 0.07 ng/mL. There are many scientific articles about fermented red beetroot juice. They concern the range of volatile or bioactive compounds of the polyphenols and red and yellow betalains. Still, none relate to the contained contaminants, in particular perfluoroalkyl substances. To the best of our knowledge, this paper is the first detailed and cross-sectional investigation of selected perfluoroalkyl substances in human bodily fluids after exposure to fermented red beetroot juice. Likewise, with red beetroot, recent modern studies have shown a variety of health benefits from it and its active compounds, and betalains (also betanin) having antioxidative, anti-inflammation, anticancer, blood pressure and lipid lowering, and also antidiabetic and anti-obesity effects. Still, it is hard to find information about its contamination, especially with PFASs. In her study, Sznajder-Karatzyńska et al. [2] investigated 55 samples of locally grown and imported fruits and vegetables. Among other things, they determined the content of 10 PFASs in red beetroot. In this case, the only identified PFAS was PFOA, and its content ranged from 0.050 to 0.090 (ng g^−1^ ww). For beetroot, the frequency of its occurrence was 100%. There is no possibility of comparing these data with the limits provided by the relevant agencies because they have not been established and do not exist. On 26 August 2022, the Commission Recommendation (EU) 2022/1431 on monitoring the presence of perfluoroalkyl substances in food was issued. Also, Herzke et al. [54] studied the contamination of vegetables by PFASs and reported that perfluorocarboxylic acids were the most frequently detected substances, with PFOA as the most abundant group. Its concentration was found to be between 0.008 and 0.121 ng g^−1^ of fresh weight. These findings are in agreement with our study.

In the case of the volunteers’ blood plasma collected before and after intake of the fermented red beetroot juice, no statistical difference was observed in the median value of the content of perfluoroalkyl acids (Table 2). Among the twenty-four volunteers tested, an upward trend in the PFOA content was observed for five people. A downward trend in the median value was observed for the other identified perfluorocarboxylic acids, PFDA and PFNA (Table 2). And for individual results, in the case of PFNA and 10 volunteers, there was a statistically significant decrease in the analyte content in the tested blood plasma samples, and for PFDA in as many as 14. The sum of the determined acids also showed a statistically significant downward trend (Table 2), which can be explained by the fact that, excluding the consumption of certain products (which was required by the assumptions of the experiment), the supply of perfluoroalkyl acids was limited. These compounds are found in significant amounts in packaging dedicated to food storage [31], thus finding their way into the human body. Reducing the consumption of processed food thus resulted in a decrease in the supply of these specific compounds in the diet.

In the case of the determined perfluoroalkane sulfonates, a statistically significant upward trend was observed (*p* < 0.01). Such a situation took place both for individually identified compounds, i.e., PFOS (14 volunteers) and PFHS (22 volunteers), and for their sum (Table 2). This slight but statistically significant increase could have been caused by the supply of other food products consumed by the volunteers.

### 2.2. Blood Parameters

To the best of the authors’ knowledge, this study is the first to show the effect of a six-week intake of fermented beetroot juice on the twelve hematology and five blood biochemistry parameters.

The blood parameter values before and after the intervention are presented in Table 3. The analysis of the hematology parameters indicated that the intake of the fermented beetroot juice showed a significant decrease in MCV, platelets concentration, MPV, P-LCR at the significance level *p* < 0.01, and hematocrit (*p* < 0.05). On the other hand, the dietary intervention also indicated a significant (*p* < 0.01) increase in corpuscular/cellular hemoglobin concentration (MCHC). In the case of blood biochemistry, no significant change in blood samples after intake of the fermented beetroot juice was observed (Table 3). However, a decreasing tendency of total cholesterol and LDL-C concentration was observed. Previously published data showed an ambiguous influence of red beet products on blood parameters.

Nevertheless, the previous studies investigated only the three blood parameters: total cholesterol, LDL-C, and HDL-C [55,56]. The study by de Castro et al. [57], conducted on volunteers (n = 36) with overweight/obesity and dyslipidemia who consumed freeze-dried red beet leaves (2.5 g/day) for four weeks, showed a significant reduction in LDL. However, in our study, as mentioned above, the concentration of LDL showed only a decreased tendency. Also, no statistical difference in the total cholesterol concentration before and after the intervention was observed in the cited study. In turn, the study by Asgary et al. [58], conducted on 24 subjects, showed a significant reduction in total cholesterol and LDL-C after the consumption of raw beetroot juice (250 mL/day) for two weeks.

On the other hand, Asgary et al. [58] noticed no significant changes in the case of these same parameters (total cholesterol and LDL-C) after the intake of cooked beetroots (250 mg/day). The blood parameters may affect the red beet product (food matrix) type, dose, and intervention time. It should be mentioned that beetroot is rich in several bioactive compounds, i.e., betalains, phenolics, carotenoids, minerals, and vitamins [55], which may affect the blood parameters tested. The dominant substances present in beetroot products are betalains. Betalains are characterized by a strong antioxidant effect, which results from the structure of their molecule. Therefore, it is primarily thanks to these compounds that beetroot is among the ten vegetables with the strongest antioxidant properties [56]. Previous studies indicate that the consumption of betalains has many positive aspects, including inhibiting lipid peroxidation, having a protective effect on red blood cells, preventing oxidative hemolysis, and having anti-carcinogenic properties [59].

Changes in blood parameters may signal adverse health effects caused by, e.g., nutritional intervention. One of the blood parameters that showed a decrease after the consumption of fermented beetroot juice was MCV. MCV is routinely measured in blood tests and is generally used to help classify the cause of anemia [60]. We observed that platelet-related parameters, such as platelet count, MPV, and P-LCR, decreased in blood samples analyzed after nutritional intervention. Generally, the platelet count determines the bleeding risk and monitors thrombopoiesis [61]. Moreover, increasing evidence describes the important role of platelets in physiological processes such as immune response, angiogenesis, and fibrosis formation [61,62]. This suggests that platelets independently influence morbidity and mortality, rather than merely reflecting underlying disease. An abnormal platelet count indicates poor prognosis in some patient groups, including in cancer and in critically ill patients [61,63,64].

The correlations between the individual PFAS compounds, total perfluorocarboxilic acids, perfluoroalkane sulfonates and PFASs and blood parameters are presented in Figure 1. Before intervention, moderate positive correlations (0.3 ≤ r < 0.5) were observed between PFHS and erythrocytes, hemoglobin, and hematocrit. Also, a moderate positive correlation was calculated between total perfluoroalkane sulfonates and erythrocytes. A good negative correlation (0.5 to <0.7) was observed between RDW-CV and PFOA, while moderate negative correlations were determined between RDW-CV and PFNA, total perfluorocarboxilic acids, and total PFASs. Additionally, moderate positive correlations were observed between glycated hemoglobin, PFOS, PFHS, and total perfluoroalkane sulfonates. Moreover, before the intervention, a positive tendency was observed between PFOS and erythrocytes and LDL-C and between total perfluoroalkane sulfonates and hemoglobin, hematocrit, and LDL-C. A negative trend was found between total perfluoroalkane sulfonates and HDL-C. On the other hand, after diet intervention, a moderate negative correlation was only observed between cholesterol HDL and PFHS (Figure 1). A negative tendency was observed between PFOS, total perfluoroalkane sulfonates, and HDL-C. Moreover, a positive tendency was noticed after the intake of the fermented beetroot juice between PFOS, total perfluoroalkane sulfonates and erythrocytes and total PFAs and hemoglobin.

However, some limitations of the study should be taken into account. First, the lack of control over food intake and potential sources of PFASs. Second, we assessed blood parameters on two points that may not account for intra-individual variability in blood parameters. Also, the number of participants participating in the study was relatively small, and could not show unambiguous statistical differences between individual parameters.

## 3. Materials and Methods

### 3.1. The Chemicals, Reagents, and Study Material

Gradient reagents, including methanol, acetonitrile, formic acid, and water, were purchased from Sigma Chemical Co. (St. Louis, MO, USA). A native standard mixture of PFASs containing seven perfluorocarboxylic acids (PFCAs) such as perfluorobutanoic acid (PFBA), perfluoropentanoic acid (PFPeA), perfluorohexanoic acid (PFHxA), perfluoroheptanoic acid (PFHpA), perfluorooctanoic acid (PFOA), perfluorononanoic acid (PFNA), and perfluorodecanoic acid (PFDA) and three perfluoroalkane sulfonates (PFSAs), namely perfluorobutane sulfonate (PFBS), perfluorohexane sulfonate (PFHxS), and perfluorooctane sulfonate (PFOS) prepared in methanol, with a chemical purity of >98% each, were purchased from Wellington Laboratories, Inc. (Guelph, ON, Canada). The isotopically labelled internal standards (ISs), perfluoro-n-[13C8] octanoic acid (13C8-PFOA) in methanol, with chemical purity of >98%, and sodium perfluoro-1-[13C8] octane sulfonate (13C8-PFOS) in methanol, with chemical purity of >98%, were obtained from Wellington Laboratories, Inc. (Guelph, ON, Canada). Auxiliary equipment such as an MPW-351R Centrifuge (MPW Med. Instruments, Warsaw, Poland), Vacuum Concentrator Plus (Eppendorf AG, Hamburg, Germany) and ultrasonicator were used for sample preparation.

Stock, intermediate, and working standard solutions of native PFASs and internal standards (13C8-PFOA and 13C8-PFOS named IS1 and IS2, respectively) were prepared in MeCN. Intermediate and working standard solutions of native PFASs with concentrations of 100 ng/mL and 1 ng/mL, respectively, were prepared by diluting the standards with a mixture of 20% MeOH in water (*v*/*v*) with the addition of 1% (*v*/*v*) of FA. Internal standard solutions were prepared according to the above procedure.

Fermented red beetroot juice was custom-prepared by a fruit and vegetable processing company in Poland.

### 3.2. Characteristic of Participants and Study Design

The subjects who met the inclusion criteria (body mass index (BMI) under 30; without gastrointestinal disturbances, including gastric and duodenal ulcers; and they could not have participated in other clinical trials within 90 days before the survey, take drugs, abuse alcohol, be pregnant and breast-feeding, or take any medications or vitamin supplements) and were certified healthy at a medical interview were accepted to the study. Ultimately, 24 healthy subjects, 5 males and 19 females, aged between 24 and 40, participated in the experiment (Table 4).

The study was conducted for 6 weeks (42 days). For 42 days, once a day, volunteers consumed a dose of the fermented beetroot juice (200 mL/60 kg of body weight) directly after breakfast. Every 7 days, the health status of all volunteers was assessed by the doctor, and they received fermented beetroot juice for the next week. Under fasting conditions, before consumption (sample zero), and at the end of the experiment, the blood samples were taken into heparinized vacutainers and then centrifuged (1000× *g*, 10 min, 4 °C). According to the above sampling scheme (0 and 42 days), urine samples were collected from the volunteers. After that, the separated plasma and collected urine were frozen and stored at 80 °C until analysis. The study design is presented in Figure 2.

### 3.3. Ethical Aspects

The experimental design and procedure were accepted by the Bioethical Committee at the Faculty of Medical Science of the University of Warmia and Mazury in Olsztyn (Poland, No. 7/2015). All volunteers were fully informed about the potential benefits and risks and signed an informed consent form. Moreover, the study was conducted under medical supervision in the NZOZ Atarax Clinic in Olsztyn, Poland. 

### 3.4. Plasma and Urine Samples Preparation

The perfluoroalkyl substances content was determined in the plasma and urine samples. The plasma samples were prepared according to Rotander et al. [65]. Briefly, 200 µL of plasma with 4 µL of ISs solution (2.5 µg mL/1) was extracted with 1.5 mL 100% acetonitrile (MeCN) using ultrasonication followed by vortex extraction, centrifugation and evaporation to dryness in a vacuum concentrator. The residue was reconstituted with 100 µL of MeOH. Before the micro-HPLC-MS/MS analysis, samples were diluted fivefold in deionised water with 1% (*v*/*v*) of FA addition to the final volume of 500 µL. The same analytical procedure was applied to blank samples. Each sample for the assay was prepared in triplicate.

The urine samples were prepared according to Perez et al. [66]. Briefly, 500 µL of urine with 3.2 µL of ISs solution (2.5 µg mL/1) was mixed. The precipitation of traces of protein was induced by mixing the samples with acetonitrile (1:1). After centrifugation at 4000 rpm for 10 min, 400 µL of the supernatant was transferred to a PP tube and evaporation to dryness in a vacuum concentrator. The residue was reconstituted with 80 µL of MeOH. Before the micro-HPLC-MS/MS analysis, samples were diluted fivefold in deionised water with 1% (*v*/*v*) of FA addition to the final volume of 400 µL. The same analytical procedure was applied to blank samples. Each sample for the assay was prepared in triplicate.

### 3.5. Instrumental Analysis

The investigated substances were analyzed using micro-HPLC/MS/MS with the negative ion electrospray ionization (ESI) and Multiple Reaction Monitoring (MRM) mode. Chromatographic separation was carried out using an Eksigent LC200 System (AB SCIEX, Concord, ON, Canada). The column used was a HALO C18 column 50 mm × 0.5 mm × 2.7 µm (Eksigent, Concord, ON, Canada), and the thermostat was set at the temperature of 45 °C with the mobile phase flow rate of 20 µL/min. A binary gradient consisting of water (A) and MeCN (B) (both with 0.1% FA) was applied. The gradient was set as follows: 40% B (0–0.5 min), 40–90% B (0.5–3.0 min), 90% B (3.0–4.0 min), 90–40% B (4.0–4.2 min), and 40% B (4.2–5.0 min). The injection volume was 5 µL. The autosampler temperature was set at 40 °C. The mass spectrometer used was QTRAP 5500 with ESI (AB SCIEX, Concord, ON, Canada). The optimal sensitivity for the investigated PFASs was obtained under the following settings: curtain gas flow, 25 L/min; collision gas flow, 9 L/min; ion spray voltage, −4500 V; temperature, 350 °C; 1 ion source gas flow, 30 L/min; 2 ion source gas flow, 35 L/min; declustering potential range, −30 to −85 V; entrance potential, −10 V; collision energy range, −10 to −65 eV; and collision cell exit potential range, −10 to −38 V [67]. Quantitative analyses were performed using the multiple reaction monitoring (MRM) mode. Data analysis was carried out with Analyst Software (AB SCIEX, Concord, ON, Canada) (version 1.5.2).

### 3.6. Analysis of Blood Parameters

The analysis of blood samples was conducted by the Medical Diagnostic Laboratory of the Provincial Specialist Hospital in Olsztyn, Poland. The blood hematology parameters included leukocytes, erythrocytes, hemoglobin, platelets, hematocrit, mean corpuscular volume (MCV), mean corpuscular hemoglobin (MCH), mean corpuscular/cellular hemoglobin concentration (MCHC), red cell distribution width (RDW-CV), platelet distribution width (PDW), mean platelet volume (MPV), and platelet large cell ratio (P-LCR), and blood biochemistry parameters were also measured, which included lipid profile (total cholesterol, triglycerides, high-density lipoprotein (HDL), low-density lipoprotein (LDL)) and glycated hemoglobin.

### 3.7. Statistical Analyses

Statistical analyses were performed using Statistica software (v. 13, StatSoft, Tulsa, OK, USA). The normal distribution of the data was evaluated using a Shapiro–Wilk W test. The tested groups showed a non-normal distribution, and therefore quantitative variables were expressed as median (P25-P75). Comparisons within the groups, between the baseline and after the exposure to the fermented red beetroot juice, were performed using Wilcoxon signed-rank tests. Correlations between the concentration of individual PFASs, total acids, total sulfonates, total PFASs, and blood parameters (leukocytes, erythrocytes, hemoglobin, hematocrit, MCV, MCH, MCHC, platelets, RDW-CV, PDW, MPV, P-LCR, total cholesterol, cholesterol HDL, cholesterol LDL, triglycerides, glycated hemoglobin) were analyzed using a Pearson correlation coefficient test. Statistical significance thresholds were set at *p* < 0.05 (*) and *p* < 0.01 (**). The strength of the correlation was described as fair (<0.3), moderate (0.3 to <0.5), good (0.5 to <0.7), or very good (≥0.7) [68].

## 4. Conclusions

This was the first time this type of experiment has been carried out, and although we cannot draw clear conclusions regarding the increase in the content of perfluoroalkyl substances in human bodily fluids due to the consumption of food contaminated with them, it proves that such research is necessary and should be continued. In the case of perfluorocarboxylic acids, a downward trend was observed, while perfluoroalkane sulfonates’ plasma content showed a statistically significant upward trend. This confirms previous scientific data showing that these compounds metabolize very poorly; therefore, they are very poorly excreted from the human body—they only accumulate in it, which makes them even more dangerous for the consumer. On the other hand, it has been observed that fermented red beetroot juice can affect blood parameters. The effect of beetroot products depends on the type of food matrix, dose, and time of intervention. It is worth mentioning that beetroot is rich in several bioactive compounds (betalains, phenols, carotenoids, minerals, and vitamins) that can have a synergistic effect on the blood parameters tested.

More research is needed to determine the sources and impact of PFASs on consumer health. Ongoing research is critical to developing future strategies to control consumer exposure to PFASs and points to areas for the further improvement of multidisciplinary collaboration. 

While preliminary, our study is among the first to explore the correlation between PFAS consumption and its potential accumulation in bodily fluids, highlighting an understudied area of consumer health.

## Figures and Tables

**Figure 1 ijms-24-13956-f001:**
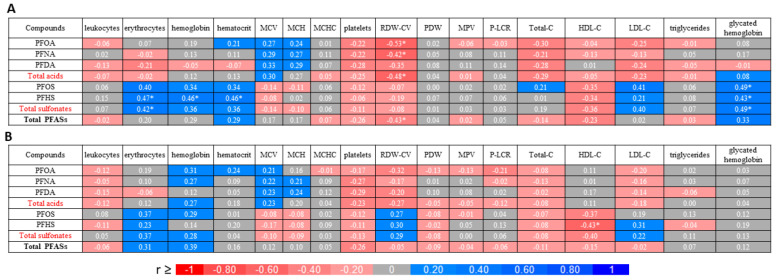
Correlation heat map between individual compounds, total acids, sulfates and PFASs and blood parameters before (**A**) and after (**B**) intake of fermented red beetroot juice. (*)—Significant correlation, (*p* < 0.05).

**Figure 2 ijms-24-13956-f002:**
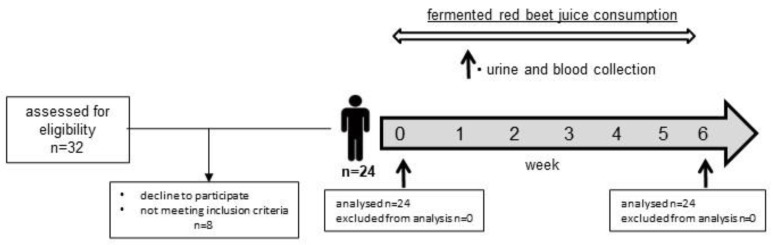
Flow diagram of participant recruitment during the study.

**Table 1 ijms-24-13956-t001:** PFASs detected in fermented red beetroot juice and the urine and blood plasma of volunteers.

No.	Compounds	R_t_ (min)	[M]^−^ (*m*/*z*)	MS/MS (*m*/*z*)	Sample
*Perfluorocarboxilic acids*					
1	PFOA	2.02	413	369	J, B, U
2	PFNA	2.43	463	419	B
3	PFDA	2.86	513	469	B
*Perfluoroalkane sulfonates*					
4	PFOS	2.97	499	80	B
5	PFHS	2.11	399	80	B

R_t_—retention time; J—red beetroot juice; B—blood plasma; U—urine.

**Table 2 ijms-24-13956-t002:** Blood plasma profile of PFASs in volunteers before (T0) and after (T1) intake of fermented red beetroot juice. Values are expressed in ng/mL and are presented as median (P25–P75).

No.	Compounds	Blood Plasma Samples		*p* T0 vs. T1
		T0	T1	
*Acids*				
1	PFOA	0.94 (0.76–1.56)	0.93 (0.77–1.52)	0.627
2	PFNA	0.39 (0.26–0.58)	0.34 (0.23–0.58)	<0.001 **
3	PFDA	0.30 (0.20–0.46)	0.23 (0.17–0.38)	<0.001 **
Total acids		1.65 (1.29–2.94)	1.49 (1.19–2.64)	0.009 **
*Sulfonates*				
4	PFOS	0.59 (0.44–0.80)	0.71 (0.53–0.96)	<0.001 **
5	PFHS	0.11 (0.09–0.17)	0.25 (0.15–0.37)	<0.001 **
Total sulfonates		0.67 (0.53–0.92)	0.96 (0.68–1.37)	<0.001 **
Total of PFASs		2.66 (1.90–4.47)	2.62 (2.08–4.17)	0.031 *

* *p* < 0.05; ** *p* < 0.01.

**Table 3 ijms-24-13956-t003:** Descriptive statistics for blood parameters before (T0) and after (T1) consuming fermented red beet juice, expressed as median (P25–P75).

Blood Parameters	Samples		*p* T0 vs. T1
	T0	T1	
*Hematology*			
Leukocytes [k/μL]	5.2 (4.9–6.0)	5.1 (4.6–5.8)	0.961
Erythrocytes [mln/μL]	4.6 (4.3–5.1)	4.7 (4.3–4.9)	0.246
Hemoglobin [g/dL]	13.2 (12.9–14.7)	13.6 (12.7–14.1)	0.721
Hematocrit [%]	41.0 (39.0–45.0)	40.0 (37.0–42.0)	0.013 *
MCV [fl]	88.0 (85.0–92.0)	87.0 (84.0–90.0)	0.001 **
MCH [pg]	29.0 (28.0–30.0)	29.0 (28.0–31.0)	0.686
MCHC [g/dL]	32.9 (32.3–33.5)	33.4 (33.1–34.2)	<0.001 **
RDW–CV [%]	13.0 (12.0–14.0)	13.0 (12.0–13.0)	0.208
Platelets [k/μL]	273.0 (215.0–318.0)	262.0 (202.0–299.0)	0.002 **
PDW [fl]	14.0 (13.0–15.0)	14.0 (12.0–15.0)	0.083
MPV [fl]	11.2 (10.7–11.9)	10.9 (10.4–11.7)	<0.001 **
P–LCR [%]	34.0 (29.0–38.0)	32.0 (28.0–38.0)	<0.001 **
*Blood biochemistry*			
Total–C [mg/dL]	184.0 (165.0–197.0)	168.0 (155.0–194.0)	0.074
HDL–C [mg/dL]	69.0 (64.0–86.0)	73.0 (57.0–88.0)	0.897
LDL–C [mg/dL]	80.6 (64.8–109.8)	76.4 (66.4–98.4)	0.128
Triglycerides [mg/dL]	80.0 (48.0–102.0)	71.0 (58.0–81.0)	0.412
Glycated hemoglobin [%]	5.2 (5.0–5.3)	5.2 (5.1–5.3)	0.877

MCV—mean corpuscular volume; MCH—mean corpuscular hemoglobin; MCHC—mean corpuscular/cellular hemoglobin concentration; RDW-CV—red cell distribution width; PDW—platelet distribution width; MPV—mean platelet volume; P-LCR—platelet large cell ratio; Total-C—total cholesterol; HDL—high density lipoprotein; LDL—low density lipoprotein; * *p* < 0.05; ** *p* < 0.01.

**Table 4 ijms-24-13956-t004:** Anthropometric parameters and demographic characteristics of the sample (% or mean and standard deviation, SD).

Characteristics	Total
Sample size	24
Sample percentage	100.0
Age (years), mean (SD)	29.5 (3.6)
Gender, n (%)	
Women	19 (79.2)
Men	5 (20.8)
BMI (kg/m^2^), mean (SD)	24.9 (1.8)
Residence, n (%)	
Urban	17 (70.8)
Rural	7 (29.2)

## Data Availability

The datasets used and/or analyzed during the current study are available from the corresponding author upon reasonable request.

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
