# Peer review of "Relationship between the Consumption of Fermented Red Beetroot Juice and Levels of Perfluoroalkyl Substances in the Human Body’s Fluids and Blood Parameters"

_ijms, 2023, doi:10.3390/ijms241813956_

Round 1

Reviewer 1 Report (New Reviewer)

The authors have presented a noteworthy study exploring the potential health impact of ingesting fermented red beetroot contaminated with perfluoroalkyl substances. Given the current search for novel nutrient sources, this study holds relevance. However, there are several significant concerns with the manuscript that need to be addressed:

1. The manuscript requires a thorough review for typographical errors, punctuation mistakes, and inconsistent sentence spacing.

2. The title is unclear. It should be rephrased: “Relationship Between Consumption of Fermented Red Beetroot Juice and Levels of Perfluoroalkyl Substances in Human Body Fluids and Blood Parameters.”

3. In the abstract, I suggest changing the word “intake” (line 17) to “ingestion.”

4. The introduction appears redundant in certain areas. Specifically, the content in Lines 74 and 75 echoes the information in Lines 101 to 103. Additionally, the sections discussing the industrial applications, health impacts, and toxicity of PFAS should be consolidated and structured more coherently.

5. Within the discussion, especially from Lines 219 to 226, the authors should specify the number of participants involved in the referenced studies for a clearer comparison.

6. From Lines 236 to 238, please identify and cite research that discusses the bioactive compounds in beetroot that may account for the observed changes in blood parameters. It would also be beneficial if the authors could highlight any clinical studies investigating the effects of these compounds or beetroot on health.

The authors demonstrate a reasonable understanding of English; however, the manuscript needs a comprehensive review for typographical errors, punctuation inaccuracies, and inconsistent spacing between sentences.

Author Response

The authors have presented a noteworthy study exploring the potential health impact of ingesting fermented red beetroot contaminated with perfluoroalkyl substances. Given the current search for novel nutrient sources, this study holds relevance. However, there are several significant concerns with the manuscript that need to be addressed:
Our answer: Thank you for a very positive assessment of our study.

  1. The manuscript requires a thorough review for typographical errors, punctuation mistakes, and inconsistent sentence spacing.
    Our answer: Thank you for this comment. The correction was done.

  2. The title is unclear. It should be rephrased: “Relationship Between Consumption of Fermented Red Beetroot Juice and Levels of Perfluoroalkyl Substances in Human Body Fluids and Blood Parameters.”
    Our answer: We agree with the comment. The correction was done.

  3. In the abstract, I suggest changing the word “intake” (line 17) to “ingestion.”
    Our answer: Thank you for this comment. The correction was done.

  4. The introduction appears redundant in certain areas. Specifically, the content in Lines 74 and 75 echoes the information in Lines 101 to 103. Additionally, the sections discussing the industrial applications, health impacts, and toxicity of PFAS should be consolidated and structured more coherently.
    Our answer: Thank you for this comment. The correction was done. The sections discussing the industrial applications, health impacts, and toxicity of PFAS were consolidated and structured more coherently.

  5. Within the discussion, especially from Lines 219 to 226, the authors should specify the number of participants involved in the referenced studies for a clearer comparison.
    Our answer: Thank you for this comment. The correction was done.

  6. From Lines 236 to 238, please identify and cite research that discusses the bioactive compounds in beetroot that may account for the observed changes in blood parameters. It would also be beneficial if the authors could highlight any clinical studies investigating the effects of these compounds or beetroot on health.

Our answer: Thank you for this comment. The additional information has been added.

Reviewer 2 Report (New Reviewer)

The manuscript titled "Exposure to the fermented red beetroot juice in relation to the level of perfluoroalkyl substances in the human body fluids and blood parameters" by Surma et al., submitted to the "International Journal of Molecular Sciences", delves into a topic of high relevance in the contemporary landscape of food science and human health. Specifically, the authors aim to elucidate the potential correlation between fermented red beetroot juice consumption and the levels of Per- and poly-fluoroalkyl substances (PFASs) in volunteers' blood plasma and urine. In addition to this, the study analyses the resultant impacts on various blood parameters. This approach is particularly interesting and timely, given the escalating concerns surrounding PFASs, especially due to their known environmental persistence and potential health implications.

After thoroughly reviewing the manuscript, I commend the authors for their efforts in addressing such a pertinent issue. However, I believe there are areas within the manuscript that can benefit from constructive feedback, which would not only enhance the overall quality of the paper but also bolster its scientific contribution to the existing body of research. Below are my specific comments and suggestions that aim to assist the authors in refining their work further:

       In the introduction, while it is essential to back claims with relevant references, the dense use of citations may interrupt the flow for readers. Consider grouping or rephrasing sentences to consolidate citations.

       Some information appears more than once. For instance, PFASs being detected in human blood, both serum and plasma, is mentioned a couple of times. It would be best to avoid such repetitions to maintain clarity and brevity.

       The introduction to PFASs may benefit from a clearer initial definition. Instead of immediately diving into their chemical structure, consider a succinct initial statement on their prevalence and concerns, followed by a detailed description.

       The way PFASs are introduced in the context of red beetroot can be misinterpreted as if red beetroot inherently contains PFASs. It may be helpful to clarify whether the concern is due to environmental contamination or some other source.

       The closing paragraph does well to set the study's objectives. However, consider expanding on why fermented red beetroot juice was specifically chosen. Is there something particular about the fermentation process that might affect PFASs content or bioavailability?

       Lines 145-146. The line "very poorly; they only accumulate in it, which makes them even more dangerous" might be redundant, as "very poorly" was already mentioned in the preceding line.

       In the results and discussion section, the authors should delve a bit into the potential implications or reasons for these changes. For instance, why might fermented beetroot juice have different effects than raw beetroot juice or freeze-dried leaves?

       Consider discussing the implications of PFASs concentrations. Are they considered high or low compared to other studies or based on health guidelines? What might be the potential health risks, if any, at these levels?

       It is always a good practice in research papers to acknowledge any study limitations. For instance, were there any potential sources of error in detecting PFASs? Were there any challenges in recording accurate blood parameters? Acknowledging these can lend credibility to your research.

       In your discussion, hint at potential avenues for future research. For instance, given the changes in blood parameters, what other areas related to beetroot juice consumption warrant exploration? Are there other substances in beetroot juice that might impact human health?

       It would be beneficial to format chemical formulas, concentrations, and specific instruments consistently.

       Instead of "Fermented red beetroot juice was specially prepared for this study by the fruit and vegetable processing company s headquartered in Poland.", perhaps specify or simplify with: "Fermented red beetroot juice was custom-prepared by a fruit and vegetable processing company in Poland."

       In the section about the blood parameters analysis, ensure that all acronyms are introduced before they are abbreviated. For example, you introduced "mean corpuscular/cellular hemoglobin concentration (MCHC)" but went straight to acronyms for others like RDW-CV.

       In the participant section, while you have detailed the inclusion criteria, you might want to mention any exclusion criteria, if they exist.

       When mentioning the centrifugation speeds (e.g., "500 × g, 15 min, 1000 × g, 10 min, 4°C"), breaking down each centrifugation step might be clearer to avoid confusion.

       The instrument settings in section 3.5 are very detailed, which is excellent for reproducibility. However, you might want to separate settings related to the instrument itself (like curtain gas flow, ion source gas flow, etc.) from those specific to the analysis of PFASs (like declustering potential range, collision energy range, etc.).

       The statistical methods used are described well. However, ensure that any assumptions made before using these tests are met (e.g., normality for Pearson correlation).

       Ensure consistency in naming and abbreviations. For instance, once you have introduced an abbreviation, like "PFASs" or "MeOH", use it consistently throughout the text.

       In the conclusions, It's vital to mention why this experiment's novelty is significant. What gap does it fill in the existing literature? You could say something like, "While preliminary, our study is among the first to explore the correlation between PFAS consumption and its potential accumulation in body fluids, highlighting an understudied area of consumer health."

Upon reviewing the provided section of the manuscript, the English language usage is generally clear and coherent. However, there are instances where sentence structuring and clarity can be enhanced. It would be beneficial to undertake a thorough proofreading for consistent verb tense usage, precise vocabulary selection, and minor grammatical nuances. Addressing these areas will further refine the manuscript and ensure that the scientific findings are presented in the most clear and compelling manner possible.

Author Response

The manuscript titled "Exposure to the fermented red beetroot juice in relation to the level of perfluoroalkyl substances in the human body fluids and blood parameters" by Surma et al., submitted to the "International Journal of Molecular Sciences", delves into a topic of high relevance in the contemporary landscape of food science and human health. Specifically, the authors aim to elucidate the potential correlation between fermented red beetroot juice consumption and the levels of Per- and poly-fluoroalkyl substances (PFASs) in volunteers' blood plasma and urine. In addition to this, the study analyses the resultant impacts on various blood parameters. This approach is particularly interesting and timely, given the escalating concerns surrounding PFASs, especially due to their known environmental persistence and potential health implications.

After thoroughly reviewing the manuscript, I commend the authors for their efforts in addressing such a pertinent issue. However, I believe there are areas within the manuscript that can benefit from constructive feedback, which would not only enhance the overall quality of the paper but also bolster its scientific contribution to the existing body of research. Below are my specific comments and suggestions that aim to assist the authors in refining their work further:

Our answer: Thank you for a very positive assessment of our study.

In the introduction, while it is essential to back claims with relevant references, the dense use of citations may interrupt the flow for readers. Consider grouping or rephrasing sentences to consolidate citations.

Our answer: Thank you for this comment. The correction was done.

Some information appears more than once. For instance, PFASs being detected in human blood, both serum and plasma, is mentioned a couple of times. It would be best to avoid such repetitions to maintain clarity and brevity.

Our answer: Thank you for this comment. The correction was done.

The introduction to PFASs may benefit from a clearer initial definition. Instead of immediately diving into their chemical structure, consider a succinct initial statement on their prevalence and concerns, followed by a detailed description.

Our answer: Thank you for this comment. The correction was done.

The way PFASs are introduced in the context of red beetroot can be misinterpreted as if red beetroot inherently contains PFASs. It may be helpful to clarify whether the concern is due to environmental contamination or some other source.

Our answer: Thank you for this comment. The additional information has been added.

The closing paragraph does well to set the study's objectives. However, consider expanding on why fermented red beetroot juice was specifically chosen. Is there something particular about the fermentation process that might affect PFASs content or bioavailability?

Our answer: The Reviewer raises an interesting concern. The authors decided to use fermented beetroot juice due to its high health-promoting properties. In this study, we wanted to observe whether, apart from compounds with beneficial properties for health, this product is a source of toxic substances and whether they can affect the biomarkers of the human body. In the case of the effect of the fermentation process on the  PFAS content and bioavailability, there is no such scientific data. We don't know the answer to this question. The subject matter is fascinating, so we plan to conduct an experiment to examine the impact of the fermentation process on the content of PFAS in fermented products.

Lines 145-146. The line "very poorly; they only accumulate in it, which makes them even more dangerous" might be redundant, as "very poorly" was already mentioned in the preceding line.

Our answer: Thank you for this comment. Unfortunately, we cannot change this sentence because it will lose its desired meaning.

In the results and discussion section, the authors should delve a bit into the potential implications or reasons for these changes. For instance, why might fermented beetroot juice have different effects than raw beetroot juice or freeze-dried leaves?

Our answer: Thank you for this comment. The additional information has been added.

Consider discussing the implications of PFASs concentrations. Are they considered high or low compared to other studies or based on health guidelines? What might be the potential health risks, if any, at these levels?

Our answer: Thank you for this comment. The health consequences of PFASs in food products are presented in the introduction. In the case of what levels these threats result from, it is not known because there are still no limits for these compounds.

It is always a good practice in research papers to acknowledge any study limitations. For instance, were there any potential sources of error in detecting PFASs? Were there any challenges in recording accurate blood parameters? Acknowledging these can lend credibility to your research.
Our answer: Limitations in PFAS , may be the cause of errors. is an insufficiently sensitive analytical method, but this does not apply to us because the equipment used is dedicated by the EFSA Panel for the determination of these compounds. Or an unqualified analyst, which also does not apply to us, may be the cause of study limitations. Moreover, in the manuscript contains information about the limitations of the study.

In your discussion, hint at potential avenues for future research. For instance, given the changes in blood parameters, what other areas related to beetroot juice consumption warrant exploration? Are there other substances in beetroot juice that might impact human health?

Our answer: Thank you for this comment. The additional information has been added.

 It would be beneficial to format chemical formulas, concentrations, and specific instruments consistently.
Our answer: Thank you for this comment. The correction was done.

Instead of "Fermented red beetroot juice was specially prepared for this study by the fruit and vegetable processing company s headquartered in Poland.", perhaps specify or simplify with: "Fermented red beetroot juice was custom-prepared by a fruit and vegetable processing company in Poland."

Our answer:  Thank you for this comment. The correction was done.

In the section about the blood parameters analysis, ensure that all acronyms are introduced before they are abbreviated. For example, you introduced "mean corpuscular/cellular hemoglobin concentration (MCHC)" but went straight to acronyms for others like RDW-CV.

Our answer:  Thank you for this comment. The correction was done.

In the participant section, while you have detailed the inclusion criteria, you might want to mention any exclusion criteria, if they exist.

Our answer:  Thank you for this comment. Exclusion criteria are also mentioned: “they could not participate in other clinical trials within 90 days before the survey, take drugs, abuse alcohol, be pregnant and breast-feeding, or take any medications or vitamin supplements”.

When mentioning the centrifugation speeds (e.g., "500 × g, 15 min, 1000 × g, 10 min, 4°C"), breaking down each centrifugation step might be clearer to avoid confusion.

Our answer: Thank you for this comment. It was our mistype. The correction was done.

The instrument settings in section 3.5 are very detailed, which is excellent for reproducibility. However, you might want to separate settings related to the instrument itself (like curtain gas flow, ion source gas flow, etc.) from those specific to the analysis of PFASs (like declustering potential range, collision energy range, etc.).

Our answer:  And that's how they're separated. There is even a sentence in lines 377-379 “The optimal sensitivity for the investigated PFASs was obtained under the following settings:…”. In addition, the parameters of the method are spelled out in accordance with the art, as written in scientific publications.

The statistical methods used are described well. However, ensure that any assumptions made before using these tests are met (e.g., normality for Pearson correlation).

Our answer: Thank you for a very positive assessment of our study. All steps in the statistical data analysis have been checked and fulfilled.

Ensure consistency in naming and abbreviations. For instance, once you have introduced an abbreviation, like "PFASs" or "MeOH", use it consistently throughout the text.

Our answer: Thank you for this comment. The correction was done.

In the conclusions, It's vital to mention why this experiment's novelty is significant. What gap does it fill in the existing literature? You could say something like, "While preliminary, our study is among the first to explore the correlation between PFAS consumption and its potential accumulation in body fluids, highlighting an understudied area of consumer health."

Our answer: Thank you for this comment. The additional information has been added.

Round 2

Reviewer 1 Report (New Reviewer)

The authors have integrated and responded appropriately to all the reviewers’ comments. The article is acceptable for publication. 

There are no issues with the English

This manuscript is a resubmission of an earlier submission. The following is a list of the peer review reports and author responses from that submission.

Round 1

Reviewer 1 Report

I reviewed the Manuscript ijms-2488570: Long-term exposure to the fermented red beetroot juice in relation to the level of perfluoroalkyl substances in the human body fluids and blood.

The argument is interesting, nevertheless I recommend for  Revise and Resubmitted after Major Revisions.

The authors in this study intend  to determine the perfluoroalkyl substances' content in fermented  red beetroot juice and human body fluids (blood plasma and urine) before and after  long-term intake of red beetroot juice.

The argument is interesting but the manuscript needs a reorganization of the text.  The abstract is poorly written, so on the Introduction. The purpose of the study is not clear because the description of the experimental design and the results is rough. Is this a toxicology study?

The authors should better define the recruited human sample, the reason for a female gender prevalence. It would be useful to insert a descriptive table of the physical characteristics and lifestyle of the subjects composing the sample. I think the authors  should have included in the experimental design the comparison with another beet juice from another company or another vegetable juice in order to confirm their findings.

Author Response

Rewiever 1

I reviewed the Manuscript ijms-2488570: Long-term exposure to the fermented red beetroot juice in relation to the level of perfluoroalkyl substances in the human body fluids and blood.

The argument is interesting, nevertheless I recommend for  Revise and Resubmitted after Major Revisions.

The authors in this study intend  to determine the perfluoroalkyl substances' content in fermented  red beetroot juice and human body fluids (blood plasma and urine) before and after  long-term intake of red beetroot juice.

The argument is interesting but the manuscript needs a reorganization of the text.  The abstract is poorly written, so on the Introduction. The purpose of the study is not clear because the description of the experimental design and the results is rough. Is this a toxicology study?

The conducted research can be classified as toxicological research because toxicology aims to analyze and determine the mode of action of toxic substances and the consequences of their ingestion by humans. In this study, we analyzed toxic compounds in juice consumed and body fluids, and an attempt was made to demonstrate the impact of toxic substances on blood parameters.

The authors should better define the recruited human sample, the reason for a female gender prevalence. It would be useful to insert a descriptive table of the physical characteristics and lifestyle of the subjects composing the sample. I think the authors  should have included in the experimental design the comparison with another beet juice from another company or another vegetable juice in order to confirm their findings.

Thank you very much for your inspirational comments and the appreciation of our work. As suggested by the reviewer, we added a table (Table 4) characterizing the research group and a figure (Figure 2) presenting the procedure during the study. The advantage of women results from the fact that more women reported to the study. Women are more courageous and open to participating in scientific research.

The reviewer draws a very interesting remark regarding comparing the obtained test results with other juices. Unfortunately, we are not able to perform this type of experiment with the consumption of a different type of juice, among others, due to the fact that the study was conducted in he past, and we no longer have the consent of the relevant committee; secondly, we do not have access to the appropriate research equipment, which significantly affects the results obtained.

Reviewer 2 Report

Thank you very much for your interesting research. Some points must be carefully revised.

1. INTRODUCTION. Line 32 and lines 37-38. Vitamins, minerals and fibre have also biological activities, so they can be considered as bioactive compounds too.

2. INTRODUCTION. Line 37. Scientific name of red beetroot should be included.

3. INTRODUCTION. Line 47. Main industrial uses of PFAs should be mentioned.

4. RESULTS AND DISCUSSION. Section 2.1. What about other sources of PFAs apart from red beetroot? They must be considered for the discussion.

5. MATERIALS AND METHODS. Section 3.2. A flux diagram describing the study design (enrolment, allocation, follow-up, analysis, number of volunteers in each phase, excluded volunteers, discontinued interventions, etc.).

6. MATERIALS AND METHODS. Section 3.2. What about the control group?

7. CONCLUSIONS. Future perspectives must be included.

Author Response

Rewiever 2

Thank you very much for your interesting research. Some points must be carefully revised.

  1. INTRODUCTION. Line 32 and lines 37-38. Vitamins, minerals and fibre have also biological activities, so they can be considered as bioactive compounds too.

Thank you for your comment you are right. The sentence has been reformulated.

  1. INTRODUCTION. Line 37. Scientific name of red beetroot should be included.

Latin name Beta vulgaris L. subsp. vulgaris was added.

  1. INTRODUCTION. Line 47. Main industrial uses of PFAs should be mentioned.

Main industrial uses of PFAs were added.

  1. RESULTS AND DISCUSSION. Section 2.1. What about other sources of PFAs apart from red beetroot? They must be considered for the discussion.

The reviewer raises a very interesting issue. Of course, PFAs are present in other foods. One of the most commonly eaten food groups is vegetables. The average content of these substances in this group of products is 0.008 and 0.121 ng g−1 of fresh weight. The Results and Discussion section has added the relevant information to the manuscript (page 4, lines: 157-178).

  1. MATERIALS AND METHODS. Section 3.2. A flux diagram describing the study design (enrolment, allocation, follow-up, analysis, number of volunteers in each phase, excluded volunteers, discontinued interventions, etc.).

Thank you for your comment. The diagram has been added (manuscript: Figure 2)

  1. MATERIALS AND METHODS. Section 3.2. What about the control group?

Thank you for your comment. There was no control group in this study. We did not create a control group because we had a "0" point - before the nutritional intervention. The study's main aim was to determine whether regular consumption of fermented juice can affect the content of PFASs in human body fluids and blood parameters.

  1. CONCLUSIONS. Future perspectives must be included.

Thank you for your comment. The additional information has been added in conclusion.

Reviewer 3 Report

After reviewing the manuscript titled: Long-term exposure to the fermented red beetroot juice in relation to the level of perfluoroalkyl substances in the human body fluids and blood parameters I have outlined some concerns and suggestions. I suggest removing "long-term" from the title since this is subjective and not really "long-term". 

I suggest rewriting the abstract. Utilize a more detailed explanation of the study, and improve clarity. Explain the context for the substances studied and make a clearer conclusion based on the results.

The introduction is poor. Highlight the importance of the work, and why it is necessary to conduct this type of research. What are the negative side effects and health implications of perfluoroalkyl substances intake? What are the metabolism and excretion mechanisms responsible for removing the substances from the body? 

You need to add more information regarding the preparation of the fermented red beetroot juice. This is the key element of the study, and I believe it is important to describe the method of preparation and some nutritional parameters. If the processing company has a standard method for making this juice, referencing or briefly describing it would also be beneficial. Discuss the content of the PFASs in the juice and the limits provided by the relevant agencies. 

Explain why was the "wash-out phase" one week long. Why was fermented beetroot juice chosen for consumption? Add this information to the manuscript. Again, highlight the importance.

Explain why specific blood parameters were chosen to be measured. What is their relevance to the study and its objectives?

For the Pearson correlation, it could be beneficial to mention if a two-tailed or one-tailed test was used, as this can influence the results. Also, the application of any corrections for multiple comparisons should be mentioned to prevent type I errors.

Section 2.1. and 2.2. Discuss or speculate on the possible mechanisms causing these changes and how these results could be used in the future. You need to discuss any limitations encountered during the study and suggest future directions for research.

The novelty of the work lies in linking dietary exposure to PFASs and blood parameters, bringing forward a new perspective to our understanding of these compounds.

Even though the work has potential the study has serious limitations:

The number of participants tested in the study is relatively small, which limits the generalizability of the findings. Studies with larger, more diverse samples are needed to confirm the results. As I understand the study does not have a control group. This should greatly improve the comparison of results, aiding in the understanding of whether observed changes were due to the intake of fermented red beetroot juice or other factors.

With relation to the number of participants, individual diet and lifestyle variations were not controlled or accounted for, which potentially influenced PFAS levels and blood parameters. The study was conducted over a short period of time, not "long". Long-term studies would provide a more comprehensive view of how continuous exposure to PFASs affects the human body.

Furthermore, the study doesn't provide insights into the direct health outcomes related to the changes observed in blood parameters. I believe that it is essential to bridge this gap to understand the real-world implications of the findings.

The structure and flow of the writing could be improved.

Author Response

Rewiever 3

After reviewing the manuscript titled: Long-term exposure to the fermented red beetroot juice in relation to the level of perfluoroalkyl substances in the human body fluids and blood parameters I have outlined some concerns and suggestions. I suggest removing "long-term" from the title since this is subjective and not really "long-term".

Thank you for your comment, the term “long-term” has been removed.

I suggest rewriting the abstract. Utilize a more detailed explanation of the study, and improve clarity. Explain the context for the substances studied and make a clearer conclusion based on the results.

Thank you for your comment, abstract was rewrited.

The introduction is poor. Highlight the importance of the work, and why it is necessary to conduct this type of research. What are the negative side effects and health implications of perfluoroalkyl substances intake? What are the metabolism and excretion mechanisms responsible for removing the substances from the body?

Thank you for your comment. The introduction was rewrited according to Reviewers comment.

You need to add more information regarding the preparation of the fermented red beetroot juice. This is the key element of the study, and I believe it is important to describe the method of preparation and some nutritional parameters. If the processing company has a standard method for making this juice, referencing or briefly describing it would also be beneficial. Discuss the content of the PFASs in the juice and the limits provided by the relevant agencies.

Explain why was the "wash-out phase" one week long. Why was fermented beetroot juice chosen for consumption? Add this information to the manuscript.

Thank you for this comment. Information on the "wash-out phase" is provided in the manuscript ("Through the first period, volunteers were on their daily diet with the elimination of red beetroot products to wash out perfluoroalkyl substances in their systems originating from this vegetable."; page 13, lines : 303-305).

Beetroot is one of the most commonly cultivated, and the products obtained from it are one of the most frequently consumed in our country. Information on this is included in the manuscript (page 1, lines: 40-48)

Explain why specific blood parameters were chosen to be measured. What is their relevance to the study and its objectives?

The study included analysis of the most frequently tested blood parameters , necessary for the proper functioning of the body, primarily red blood cells (erythrocytes), white blood cells (leukocytes), and platelets. In our study, 17 different blood parameters were determined, thanks to which we can obtain information not only on the number of individual components but also based on which we can assess quantitatively and qualitatively different systems. Determining many different parameters in the blood made it possible to decide better whether the tested substances affect individual parameters. One parameter may not be affected by a given substance, while the other may be affected. Therefore it is crucial to test many different parameters in this type of research, considering both biochemical and hematological blood parameters. Our study observed a positive trend after intake of the fermented beetroot juice between PFOS, total perfluoroalkane sulfonates and erythrocytes and total PFAs and hemoglobins.

For the Pearson correlation, it could be beneficial to mention if a two-tailed or one-tailed test was used, as this can influence the results. Also, the application of any corrections for multiple comparisons should be mentioned to prevent type I errors.

The reviewer raises a very interesting issue. Thank you for this comment. Unfortunately, in the statistical analysis software used, we cannot choose a two-tailed or one-tailed test. However, given that the direction of the linkage was not known in advance, it may suggest that the software performed a two-tailed test.

Section 2.1. and 2.2. Discuss or speculate on the possible mechanisms causing these changes and how these results could be used in the future. You need to discuss any limitations encountered during the study and suggest future directions for research.

Thank you for your comment, abstract was rewrited. The information has been added in the conclusion.

Even though the work has potential the study has serious limitations:

The number of participants tested in the study is relatively small, which limits the generalizability of the findings. Studies with larger, more diverse samples are needed to confirm the results. As I understand the study does not have a control group. This should greatly improve the comparison of results, aiding in the understanding of whether observed changes were due to the intake of fermented red beetroot juice or other factors.

Thank you for your comment. There was no control group in this study. We did not create a control group because we had a "0" point - before the nutritional intervention. The study's main aim was to determine whether regular consumption of fermented juice can affect the content of PFASs in human body fluids and blood parameters.

It is difficult to assess whether the changes caused resulted from the consumption of beetroot juice or were influenced by other factors. Nevertheless, the statistical analysis showed a positive trend after intake of the fermented beetroot juice between PFOS, total perfluoroalkane sulfonates and erythrocytes and total PFAs and hemoglobins. The obtained results indicate the tendency and influence between the content of the tested substances and the specified blood parameters.

With relation to the number of participants, individual diet and lifestyle variations were not controlled or accounted for, which potentially influenced PFAS levels and blood parameters. The study was conducted over a short period of time, not "long". Long-term studies would provide a more comprehensive view of how continuous exposure to PFASs affects the human body.

The term “long-term” has been removed.

Furthermore, the study doesn't provide insights into the direct health outcomes related to the changes observed in blood parameters. I believe that it is essential to bridge this gap to understand the real-world implications of the findings.

Thank you for this comment, additional information has been added. (manuscript: page 6, lines: 239-250).

Round 2

Reviewer 3 Report

Line 195 - remove "long-term".

In Table 2. deviations seem very high, for example, "PFNA  values are 0.39 (0.26-0.58) at T0 and 0.34 (0.23-0.58) at T1 and p is < 0.001**. This cannot be true for the number of samples you had. You must recheck this.

Response to the comment: Thank you for this comment. Information on the "wash-out phase" is provided in the manuscript ("Through the first period, volunteers were on their daily diet with the elimination of red beetroot products to wash out perfluoroalkyl substances in their systems originating from this vegetable."; page 13, lines : 303-305).

R2: But why wash-out-period was not prolonged, e.g. blood sample analysis after one week and then blood sample analysis after two weeks to eliminate and observe whether there was the possibility of a spontaneous increase in perfluoroalkyl substances (i.e. intake from random foods)? This seems like a crucial step before the research conduction.

Moderate editing of English language required to improve clarity.